# Evidence of Maternal Antibodies Elicited by COVID-19 Vaccination in Amniotic Fluid: Report of Two Cases in Italy

**DOI:** 10.3390/v14071592

**Published:** 2022-07-21

**Authors:** Francesca Colavita, Alessandra Oliva, Aurora Bettini, Andrea Antinori, Enrico Girardi, Concetta Castilletti, Francesco Vaia, Giuseppina Liuzzi

**Affiliations:** National Institute for Infectious Diseases “L. Spallanzani” IRCCS, 00149 Rome, Italy; francesca.colavita@inmi.it (F.C.); alessandra.oliva@inmi.it (A.O.); aurora.bettini@inmi.it (A.B.); andrea.antinori@inmi.it (A.A.); enrico.girardi@inmi.it (E.G.); concetta.castilletti@inmi.it (C.C.); francesco.vaia@inmi.it (F.V.)

**Keywords:** COVID-19, pregnancy, vaccination, antibodies, passive immunity, amniotic fluid

## Abstract

With SARS-CoV-2 infection, pregnant women may be at a high risk of severe disease and adverse perinatal outcomes. A COVID-19 vaccination campaign represents the key strategy to combat the pandemic; however, public acceptance of maternal immunization has to be improved, which may be achieved by highlighting the promising mechanism of passive immunity as a strategy for protecting newborns against SARS-CoV-2 infection. We tested the anti-SARS-CoV-2 antibody response following COVID-19 full-dose vaccination in the serum and amniotic fluid of two pregnant women who presented between April and June 2021, at the Center for the Treatment and Prevention of Infections in Pregnancy of the National Institute for Infectious Diseases “L. Spallanzani”, for antenatal consultancy. Anti-SARS-CoV-2 IgG was found in residual samples of amniotic fluid collected from both women at the 18th week of gestation (63 and 131 days after the second dose’s administration). Titers in amniotic fluid mirrored the levels detected in serum and were inversely linked to the time from vaccination. Our results suggest that antibodies elicited by COVID-19 vaccination can cross the placenta and reach the fetus; therefore, they may offer passive immunity at birth. It is critical to fully understand the kinetics of the maternal response to vaccination, the efficiency of IgG transfer, and the persistence of antibodies in infants to optimize maternal immunization regimens.

## 1. Introduction

Severe acute respiratory syndrome coronavirus 2 (SARS-CoV-2) infection may pose a high risk to pregnant women of severe/critical COVID-19 and pregnancy complications [1,2]. Randomized controlled clinical trials for mRNA-based vaccines against COVID-19 excluded pregnant women, and hesitancy and reluctance still limit the vaccine coverage in this population. Growing studies show that vaccination during pregnancy is safe, immunogenic, and advantageous for both maternal and newborn health [3,4]. Maternal immunity during pregnancy has an important impact on the early protection of newborns from infectious diseases [5,6,7,8]. Previous studies showed that antibodies produced by mothers infected with SARS-CoV-2 pass through the placenta to the fetus [9]. Transplacental transfer of antibodies is a mechanism well-documented for vaccine-preventable infectious diseases. In fact, vaccination during pregnancy increases maternal antibody levels and enhances passive immunity against pathogens, representing a promising strategy for protecting newborns against infections during gestation and in the first months of life [5,10]. Here, we present two cases with evidence of maternal anti-SARS-CoV-2 IgG in amniotic fluid samples following COVID-19 vaccination at the National Institute for Infectious Diseases “L. Spallanzani” (INMI, Rome, Italy).

## 2. Materials and Methods

### 2.1. Sample Collection

Two pregnant women, vaccinated with an mRNA COVID-19 vaccine (BNT162b2, Pfizer-BioNTech), presented between April and June 2021 to the Prenatal Clinical Unit at the INMI for prenatal follow-up. Blood samples were collected at each visit to monitor the response to COVID-19 vaccination. According to the current guidelines in Italy, a TORCHES investigation was performed on both women. Due to positive results (IgM+ and IgG+ with low IgG avidity) in our serological screening for toxoplasmosis indicative of primary maternal infection, both women were prescribed amniocentesis at the 18th week of pregnancy, aimed at evaluating the risk of toxoplasmosis vertical transmission. Residual amniotic fluid samples were stored at −20 °C and subsequently tested for an anti-SARS-CoV-2 antibody response.

### 2.2. Serological Testing

The anti-SARS-CoV-2 IgG levels in residual amniotic fluid samples were evaluated using a home-made indirect immunofluorescence assay (IFA) as described elsewhere [11]. Briefly, the slides were prepared with Vero E6 cells infected with a SARS-CoV-2 isolate (2019-nCoV/Italy-INMI1, GISAID accession number EPI_ISL_410546), and specific IgG in the amniotic fluid was evaluated by twofold titration, starting from a 1:2 dilution. The levels of anti-spike/receptor binding domain (RBD) IgG were determined in serum by an automated chemiluminescent immunoassay (ARCHITECT SARS-CoV-2 IgG II Quantitative, Abbott Laboratories, Wiesbaden, Germany). According to the manufacturer’s instructions, the levels of anti-S/RBD are expressed as binding antibody units (BAU)/mL (conversion factor: 1 BAU/mL = 0.142 × AU/mL), and the linear range spans from 7.1 (positivity threshold) to 5680 BAU/mL, expanded to 11,360 with an automated dilution.

## 3. Results

The first woman (Patient A) was 37 years old and received the full schedule of COVID-19 vaccination during her pregnancy (gestational age: first dose, 7th week; second dose, 10th week). The anti-S/RBD IgG levels in serum were 474.2 and 144.4 BAU/mL at 36 (15th week of pregnancy) and 92 (23rd week of pregnancy) days after the second dose’s administration, respectively. Anti-SARS CoV-2 IgG was detected in amniotic fluid collected at the 18th week of gestation (63 days after the second dose’s administration), with a titer of 1:8 (Table 1).

The second woman (Patient B) was 39 years old and completed the COVID-19 vaccination course before her pregnancy (second dose administered on the day of the last menstrual period). Amniocentesis was performed on the 18th week of pregnancy (131 days after the second dose’s administration), and simultaneously, a blood sample was collected. Specific antibodies were detected both in serum (anti-S/RBD IgG, 92.5 BAU/mL) and amniotic fluid (anti-SARS-CoV-2 IgG, 1:2) (Table 1).

No previous infection or exposure to SARS-CoV-2 was reported for either woman, and neither had an adverse effect to the vaccination. Notably, no toxoplasma DNA was detected in either amniotic fluid sample.

## 4. Discussion

The transfer of IgG from mother to fetus across the placenta is critical to protect infants during the first few months of life. Maternal vaccination during pregnancy provides this passive immunity [5,8,12]. Several studies reported anti-SARS-CoV-2 vaccine-elicited antibodies in umbilical cord blood and breastmilk after maternal vaccination [13,14,15]. To our knowledge, this is the first description of maternal antibodies directly in the amniotic fluid elicited by the COVID-19 vaccine during gestation. The antibody levels in the amniotic fluid were lower than those found in the mother’s serum, and we observed that in these two cases, the titers found in the amniotic fluid mirrored the levels in serum (higher titer in the amniotic fluid sample of the patient with a higher level of IgG in serum) and were inversely linked to the time from vaccination (higher titer in the amniotic fluid of the patient who received the vaccination more recently).

Our observation supports the evidence that COVID-19 vaccination of pregnant women leads to transplacental antibody transfer, potentially able to protect the fetus from infection [5]. As for other vaccine-preventable diseases, maternal immunization against SARS-CoV-2 may represent an important strategy to protect infants for whom there is currently no licensed vaccine.

Passive immunity depends on multiple factors, including maternal specific antibody levels and immune response dynamics, as well as the timing of vaccination for the IgG transfer across the placenta during gestation [8]. Full maternal immunization during pregnancy has been shown to maximize transplacental antibody transfer, with adequate seroprotection in young infants [13]. Protective antibodies elicited by maternal COVID-19 vaccination in newborns’ cord blood persisted in infants for at least six months [14,15]. In addition, post-natal immunity through breast milk was evidenced following COVID-19 vaccination of lactating women; in fact, vaccine-induced anti-RBD antibodies were detected both in women’s breast milk and infant stool samples [6,16]. Overall, the beneficial ante- and postnatal effect reinforces the recommendation for COVID-19 vaccination during pregnancy. However, the question obstetricians and gynecologists may have to answer is: when should a vaccination schedule be recommended for pregnant women? The time of vaccination may be critical for maternal immunity and the efficient transfer of antibodies to the newborn. For instance, early compared to late third-trimester maternal SARS-CoV-2 full-immunization has been shown to enhance transplacental antibody transfer and increase neonatal neutralizing antibody levels [4,17]; on the other hand, the levels of the mother’s antibodies, which correlate with antenatal immunity, may drop throughout pregnancy according to the time from vaccination, with a potential benefit of a third vaccine dose for women vaccinated early in their pregnancy [4,18]. Data are still limited, and with the continuous emergence of new immune-escaping viral variants, it is critical to determine the optimal timing for maternal COVID-19 vaccination to maximize the duration of vaccine-induced antibodies and the benefit to mothers’ and infants’ immunity. For this reason, further studies on larger populations are needed to fully understand the kinetics of the maternal response to vaccination, the efficiency of IgG transfer across the placenta during gestation, and the half-life and persistence of antibodies in infants.

## Figures and Tables

**Table 1 viruses-14-01592-t001:** Maternal antibody levels in serum and amniotic fluid after second dose of COVID-19 vaccination.

Patient	COVID-19 Full Vaccination	Serum			Amniotic Fluid		
		Gestational Age (Weeks)	Days from 2nd Dose of Vaccination	Antibodies (Anti-Spike RBD IgG) ^§^	Gestational Age (Week)	Days from 2nd Dose of Vaccination	Antibodies (Anti-SARS-CoV-2 IgG) ^#^
**A**	10th week of pregnancy	15	36	474.2	18	63	1:8
**B**	Last menstrual period	18	131	92.5	18	131	1:2

Results are expressed in ^§^ BAU/mL and ^#^ reciprocal dilution.

## Data Availability

The data used in the current study are available, only for sections not infringing personal information, from the corresponding author on reasonable request.

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
