# Peer review of "Evidence of Maternal Antibodies Elicited by COVID-19 Vaccination in Amniotic Fluid: Report of Two Cases in Italy"

_viruses, 2022, doi:10.3390/v14071592_

Round 1
Reviewer 1 Report
I appreciate the originality of the topic.
Concerning the bibliography, reference nr 1 is wrongly cited as it does not approach the evolution and complication of the Covid 19 in pregnancy but it is a systematic review about safety of vaccine in pregnancy. I suggest others papers as
Di Mascio D, Khalil A, Saccone G. Outcome of coronavirus spectrum infections (SARS, MERS, COVID-19) during pregnancy: a systematic review and meta-analysis. Am J Obstet Gynecol MFM. 2020;2
D'Antonio F, Sen C, Mascio DD, Galindo A, ; On the behalf of the World Association of Perinatal Medicine working group on coronavirus disease 2019. Maternal and perinatal outcomes in high compared to low risk pregnancies complicated by severe acute respiratory syndrome coronavirus 2 infection (phase 2): the World Association of Perinatal Medicine working group on coronavirus disease 2019. Am J Obstet Gynecol MFM. 2021 Jul;3(4):100329. doi: 10.1016/j.ajogmf.2021.100329. Epub 2021 Feb 20. PMID: 33621713; PMCID: PMC7896113.
I suggest to mention in the title that there is a report of 2 cases not a study.
Author Response
- I appreciate the originality of the topic.
ANSWER: We are glad that the Reviewer appreciated our work.
- Concerning the bibliography, reference nr 1 is wrongly cited as it does not approach the evolution and complication of the Covid 19 in pregnancy but it is a systematic review about safety of vaccine in pregnancy. I suggest others papers as
Di Mascio D, Khalil A, Saccone G. Outcome of coronavirus spectrum infections (SARS, MERS, COVID-19) during pregnancy: a systematic review and meta-analysis. Am J Obstet Gynecol MFM. 2020;2
D'Antonio F, Sen C, Mascio DD, Galindo A, ; On the behalf of the World Association of Perinatal Medicine working group on coronavirus disease 2019. Maternal and perinatal outcomes in high compared to low risk pregnancies complicated by severe acute respiratory syndrome coronavirus 2 infection (phase 2): the World Association of Perinatal Medicine working group on coronavirus disease 2019. Am J Obstet Gynecol MFM. 2021 Jul;3(4):100329. doi: 10.1016/j.ajogmf.2021.100329. Epub 2021 Feb 20. PMID: 33621713; PMCID: PMC7896113.
ANSWER: We have included the two suggested references and cited the previous nr 1 for the COVID-19 vaccination description.
- I suggest to mention in the title that there is a report of 2 cases not a study.
ANSWER: We have modified the title as suggested.
Reviewer 2 Report
This is an interesting report on 2 cases of pregnant women who had received COVID vaccine, and subsequently showed that antibody was detected in the amniotic fluid. This suggest that vaccination of pregnant women may confer protection to their babies.
State the reason of risk of toxoplasmosis in these cases.
State the justification of collection of amniotic fluid for the analysis of toxoplasmosis.
Why serology testing of toxoplasma is not sufficient?
This sentence is not clear (see below). Are the days after 2nd dose and the gestational week appropriate?
36 days – 23rd week
92 days – 15th week
Anti-S/RBD IgG in serum were 474.2 BAU/mL and 144.4 BAU/mL at 36 (23rd week of pregnancy) and 92 (15th week of pregnancy) days after the second dose administration.
The normal range of antibody (SAR-CoV2 IgG) should be provided.
A complete obstetrics history of both mothers should be provided.
Is TORCHES screening performed?
I recommend to include this article (see below) as a reference.
Male V. (2022). SARS-CoV-2 infection and COVID-19 vaccination in pregnancy. Nature reviews. Immunology, 22(5), 277–282.
Author Response
- This is an interesting report on 2 cases of pregnant women who had received COVID vaccine, and subsequently showed that antibody was detected in the amniotic fluid. This suggest that vaccination of pregnant women may confer protection to their babies.
ANSWER: We are glad that the Reviewer appreciated our work.
- State the reason of risk of toxoplasmosis in these cases.
ANSWER: In Italy, in accordance with the current guidelines, amniocentesis is recommended in cases where maternal serology does not exclude a recent primary infection and is used to predict congenital Toxoplasmosis prenatally. The two patients presented positive serology for Toxoplasmosis (both IgG and IgM +) with low IgG avidity, which is suspected for infection during the first trimester with a risk of vertical transmission. We included a statement to better explain this aspect in the Materials and Methods section (Sample Collection).
- State the justification of collection of amniotic fluid for the analysis of toxoplasmosis.
ANSWER: Collection of amniotic fluid for PCR testing is crucial to verify vertical transmission and is essential for further treatment management of both mother and fetus. See also previous answer.
-Why serology testing of toxoplasma is not sufficient?
ANSWER: Serology screening for pregnant women in the first trimester is used for diagnosis of maternal infection, while amniocentesis is recommended after the 18th week of gestation in case of positive maternal serology, to assess vertical transmission and exclude risk of congenital Toxoplasmosis. See answers above.
- This sentence is not clear (see below). Are the days after 2nd dose and the gestational week appropriate?
36 days – 23rd week
92 days – 15th week
Anti-S/RBD IgG in serum were 474.2 BAU/mL and 144.4 BAU/mL at 36 (23rd week of pregnancy) and 92 (15th week of pregnancy) days after the second dose administration.
ANSWER: We thank the Reviewer, we made a mistake in transcribing the data. We corrected the table and the text.
- The normal range of antibody (SAR-CoV2 IgG) should be provided.
ANSWER: The linear range of the anti-S/RBD IgG test span from 7.1 (positivity threshold) to 5,680, expanded to 11,360 with an automated dilution. Assay linearity throughout this range has been confirmed by an independent evaluation published in Narasimhan, M, et al. J Clin Microbiol 2021;59:e0038821. https://doi.org/10.1128/jcm.00388-21. We mentioned the complete range in the method.
- A complete obstetrics history of both mothers should be provided. Is TORCHES screening performed?
ANSWER: According to the guidelines in Italy, TORCHES was performed in both women and during the first trimester, serological tests evidenced both IgM and IgG positive for Toxoplasmosis, with low avidity. No Toxoplasma DNA was detected in both amniotic fluid samples. No other serological positivity was found within TORCHES tests.
- I recommend to include this article (see below) as a reference.
Male V. (2022). SARS-CoV-2 infection and COVID-19 vaccination in pregnancy. Nature reviews. Immunology, 22(5), 277–282.
ANSWER: We have included the suggested reference (da aggiustare numerazione)